# Systolic Array Acceleration of Spiking Neural Networks with Application-Independent Split-Time Temporal Coding

## Abstract

Spiking Neural Networks (SNNs) are brain-inspired computing models with event-driven based low-power operations and unique temporal dynamics. However, spatial and temporal dynamics in SNNs pose a significant overhead in accelerating neural computations and limit the computing capabilities of neuromorphic accelerators. Especially, unstructured sparsity emergent in both space and time, i.e., across neurons and time points, and iterative computations across time points cause a primary bottleneck in data movement.

In this work, we propose a novel technique and architecture that allow the exploitation of temporal information compression with structured sparsity and parallelism across time, and significantly improves data movement on a systolic array. We split a full range of temporal domain into several time windows (TWs) where a TW packs multiple time points, and encode the temporal information in each TW with Split-Time Temporal coding (STT) by limiting the number of spikes within a TW up to one. STT enables sparsification and structurization of irregular firing activities and dramatically reduces computational overhead while delivering competitive classification accuracy without a huge drop. To further improve the data reuse, we propose an Integration Through Time (ITT) technique that processes integration steps across different TWs in parallel with a systolic array. The proposed architecture with STT and ITT offers an application-independent solution for spike-based models across various types of layers and networks. The proposed architecture delivers 77X and 60X latency and energy efficiency improvements for different benchmarks on average over a conventional SNN baseline.

## 1 Introduction

Non-spiking artificial neural networks (ANNs) process information with continuous-valued signals representing averaged firing rates of neurons resulting from activation functions such as rectified linear unit (ReLU) and sigmoid Agarap (2018); Li & Yuan (2017). In contrast, spiking neural networks (SNNs) handles unraveled information in space and time, i.e., across different neurons (space) and different time points (time), with explicitly modeled all-or-none firing spikes. As reported in recent studies, spatial and temporal dynamics with biologically inspired Kheradpisheh et al. (2018); Hao et al. (2020) and backpropagation based Park et al. (2020); Zhang & Li (2020); Jin et al. (2018) SNN training algorithms have demonstrated competitive performances for various tasks.

From a hardware acceleration point of view, SNNs have considered better positioned for low-power operations than ANNs with biologically plausible computing models including event-driven processing and binary-valued signals. However, computations along the temporal dimension and unstructured sparsity in both space and time complicate the hardware acceleration of spike-based models. The unique temporal dimension in SNNs offers an opportunity in processing complex spatiotemporal data but introduces iterative and unstructured data movement at each time point.

The two most well-known commercial neuromorphic chips, IBM's TrueNorth Akopyan et al. (2015) and Intel's Loihi Davies et al. (2018) are based on multi-core architecture and asynchronous core-to-core communication, emulating 256 spiking neurons and 1024 spiking neural units in each core, respectively. TrueNorth and Loihi achieved low-power and high performance where weights are

fully stored on-chip and executing the computations sequentially, time point by time point. While both architectures show the promise of neuromorphic computing, we recognize a critical disadvantage: iterative and unstructured data movement. Processing the computations at each time point involves data movement associated with the firing neurons, repeated across time points with unstructured firing sparsity in a pre-synaptic layer. Stereotypical approaches to process spiking neurons in a time-sequential manner degrade the computational capabilities due to significant overhead in data movement and limited data reuse across time points. As networks are becoming deeper and larger, hardware acceleration of spiking neural computations is even more memory-bounded and the above issue significantly degrades the achievable accelerator performance in many practical cases.

Moreover, the data movements are more complicated than the feedforward counterparts when recurrence is added into the network as in recurrent SNNs (R-SNNs) due to more complex spatiotemporal dynamics and tightly coupled data dependencies. R-SNNs more closely resemble cognitive processes in the human brain and have shown state-of-the-art performances in various sequential learning tasks with temporal memory to store past information. However, recurrence in network connectivity requires the information of the previous time point, which establishes a strong causal chain and introduces challenges in hardware acceleration of R-SNNs. Furthermore, accelerating R-SNNs requires alternating access to two different types of weight matrices for every time point, i.e., feedforward and recurrent weight matrix, introducing more difficulties in data reuse and minimizing the data movement.

This work aims to develop a systolic array-based architecture to tap the full potential of SNN acceleration with key techniques below:

**Split-Time Temporal coding (STT)** : We propose a novel, universally applicable solution for sparsification and structurization of any rate-based spiking activities and explore the impact of temporal granularity defined by the time window (TW) size. STT significantly improves accelerator performance by reducing the spike redundancy on a TW basis and handling the TW as the basic unit of operation with structured firing activities across TWs.

**Integration Through Time (ITT)** : ITT enables parallel acceleration in time based on simultaneous processing of multiple TWs across columns of the systolic array. ITT enables the data reuse across TWs with uniform processing times for TWs, leading to further improved performance on top of STT.

**Systolic array-based Architecture** : We develop a systolic array-based architecture supporting STT and ITT. The proposed architecture is capable of accelerating various types of layers. We overcome the causality and tightly coupled dependencies by using the prefix sum without additional resources.

We evaluate the proposed architecture and techniques with an architecture simulator based on the actual spiking activities of well-trained networks on various networks including fully-connected, convolutional and recurrent. The proposed architecture delivers 97X latency and 78X energy efficiency improvements on average over a conventional SNN baseline on different benchmarks.

## 2 SPLIT-TIME TEMPORAL CODING (STT)

The essence of STT is to split-and-structurize the spiking activities by imposing regularity in terms of the number of spikes per synchronized TW and to enable a tunable tradeoff between machine learning and accelerator performance. By creating regularized spike trains throughout the network, STT offers multiple benefits including reduced computational/data movement overhead, uniform processing time across TWs, and avoiding the processing of redundant spikes.

### 2.1 PROPOSED STT

We propose a novel technique to locally employ coding and sparisification by dividing the time stride (TS) with a temporal granularity defined by the time window (TW) size, dubbed *Split-Time Temporal coding* (STT). The key idea here is to employ local structurization and sparsification and improve the computational/data movement overhead by reducing the redundancy in locally rate-coded firing activities on a TW basis while retaining local rate information by using prefix sum. Importantly, STT is universally applicable for accelerating spiking models, with flexibility in choosing the TW size. The spike timing of the single spike coded for each TW carries firing rate information with time-

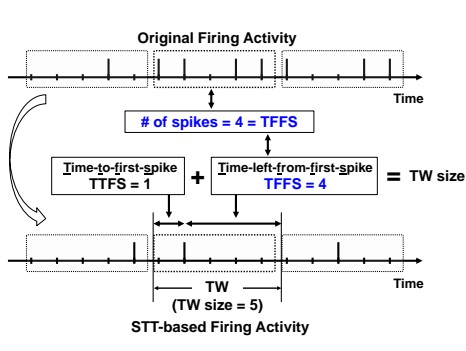

(a) Time-left-from-first-spike (TFFS)

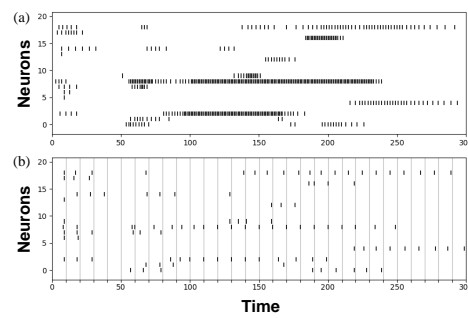

(b) a) Original firing activities without using STT and b) STT-based firing activities with TW size = 10.

Figure 1: (a) Local structurization and sparsification with the proposed STT. Time-left-from-first-spike (TFFS) presents the firing rate of the corresponding TW. (b) Spike raster plot of 20 neurons from the recurrent layer for accelerating NTIDIGITS.

left-from-first-spike (TFFS), as shown in Fig. 1(a). All layers in the network operate on TW-based local coding based on the proposed STT, with the following rules:

**Rule 1.** We limit the maximum firing count of each neuron in a TW to one. In all TWs, each neuron is allowed to fire up to once where the only spike represents rate information.

**Rule 2.** The spike count within a TW is represented by the timing of a single spike. As such for the input layer, the spike information of original input firing activity is converted with STT based on the number of spikes in each TW.

**Rule 3.** At the output layer, STT-based firing activities are decoded to firing rate. The firing rate of each neuron is decided by integrating its firing rates from all TWs, i.e., summing up all TFFS in the time domain.

As shown in Fig. 2(a), we first convert the rate-coded original firing activities into STT-based firing activities at the input layer. For example, as in Fig. 1(a), if the TW size is 5 and the number of spikes in a TW is 4, the time-left-from-first-spike (TFFS) in the corresponding TW is determined by: TFFS = (TW size) - TTFS = 5 - 1 = 4, representing the firing rate of the TW. If a neuron does not fire in a specific TW in original firing activities, the STT-based firing activity of the corresponding TW of the neuron also remains silent. In all layers in the network, each neuron follows **Rule 1** and fires at most once for a TW. For each TW throughout the layers, the timing of a single spike represent the spike information of a TW similar to Park et al. (2020). The earlier the spike, the stronger the stimulus. At the output layer, STT-based firing activities are decoded to firing rate for the decision making. For example, the spike train in Fig. 2(c) is decoded by integrating rates across TWs: $\sum$(TFFS) = $\sum$ (TW size)-(TFFS) = 2 + 4 + 1 = 7, following **Rule 3**.

## 2.2 STT-BASED ACCELERATION

STT-based hardware acceleration significantly simplifies the synaptic input integration step, the dominant computational complexity in spiking neural computations, with structured, high sparsity as shown in Fig. 1(b). First, STT reduces the repeated weight access across multiple time points to a single weight access per input neuron for a given TW. Since an input neuron fires up to once in a TW, the corresponding weight is used only once for the synaptic input integration.

Second, to retain the local and also global information, we use the prefix sum of the STT-based integrated synaptic inputs in a TW while this efficient process is still based on a single spike per TW, following **Rule 2**. As will be shown in Fig. 4 and discussed in Section 3, the prefix sum of STT-based integrated inputs is equivalent to the Psums using a left-aligned rate code where the firing rate corresponds to TFFS.

Finally, STT allows parallel acceleration through time via using a small amount of memory for each time point of a TW. For the case in Fig. 2(b), conventionally, the input integration step requires accessing weight data $W_A$ and $W_C$ at time point $t_k$, and $W_A$, $W_B$ and $W_C$ at the next time point

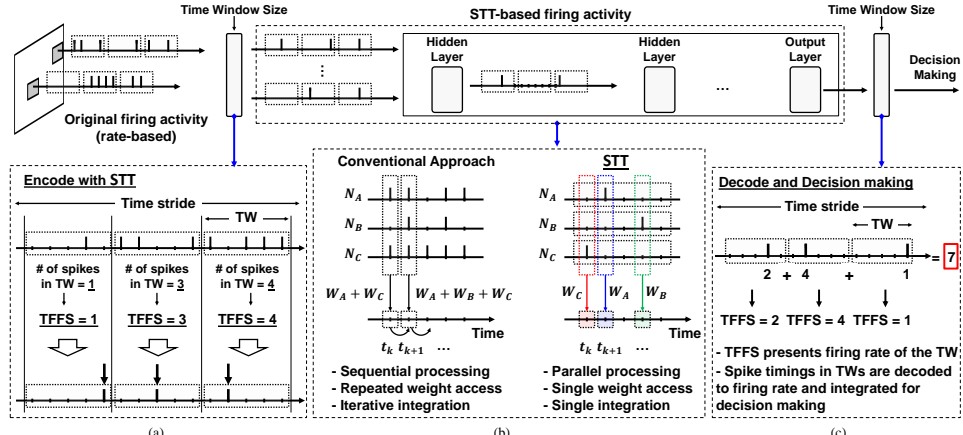

Figure 2: Schematic representations of STT-based network operations. (a): STT-encoder at the input layer (b): Comparison between the operations in conventional approaches and the proposed STT-based approach (c): STT-decoder at the output layer

$t_{k+1}$ sequentially. These unstructured firing patterns across different neurons and time points render repeated weight access without data reuse. Differently, with STT, $W_A$ is integrated to the partial sums (Psums) at $t_{k+1}$, $W_B$ is integrated to the Psums at $t_{k+3}$, and $W_C$ is integrated to the Psums at $t_k$ in parallel. Additionally, regularized spike trains, i.e., single spike per TW, makes the processing time of TWs uniform. Each weight is used only once in a TW, and multiple TWs are mapped on a systolic array simultaneously to maximize weight reuse across TWs.

# 3 PROPOSED ARCHITECTURE

We present a systolic array-based SNN accelerator architecture that supports the proposed STT and exploits parallelism in both space and time. The proposed architecture addresses existing inefficiencies via structured sparse firing patterns and parallel computations across TWs based upon the STT. In the rest of the paper, we primarily focus on synaptic input integration, the dominant computational complexity. Also, detailed overview of the proposed architecture is described in Appendix B, advantages of the proposed techniques will be discussed in Section 4 and Appendix C.

## 3.1 INTEGRATION THROUGH-TIME (ITT)

2-D systolic arrays naturally exploit parallelism and data reuse in both vertical and horizontal directions. To fully utilize such advantages, we propose an *Integration Through-Time* (ITT) technique on top of STT, which defines a spiking activity in a TW as a basic unit of workload and maps spiking activities in multiple TWs onto the systolic array, concurrently. As shown in Fig. 3(c), ITT assigns entire spike trains in a TW to a single PE and accelerates multiple TWs in different PEs simultaneously. ITT allows for accelerating multiple time points in several TWs in parallel based on the fact that the synaptic input integration step (Step 1) only depends on the spike inputs from the previous layer (2). Integration of synaptic inputs across multiple time points with ITT can be expressed by modifying (2) as:

**Step 1 - ITT:** Synaptic input integration in $TW_n \sim TW_{n+m}$:

$$
\begin{aligned}
&\mathbf{p}^{Post}[TW_n, TW_{n+1}, ..., TW_{n+m}] \\
&= \mathbf{p}^{Post}[(t_{k(n-1)+1}, ..., t_{kn}), ..., (t_{k(n+m-1)+1}, ..., t_{k(n+m)})] \\
&= \mathbf{W}_{Post,Pre} \times \mathbf{s}^{Pre}[TW_n, TW_{n+1}, ..., TW_{n+m}] \\
&= \mathbf{W}_{Post,Pre} \times \mathbf{s}^{Pre}[(t_{k(n-1)+1}, ..., t_{kn}), ..., (..., t_{k(n+m)})]
\end{aligned}
\tag{1}
$$

where $k$ is the size of the TW, $\mathbf{p}^{Post}$ and $\mathbf{s}^{Pre}$ are now matrices, and synaptic input integration is processed across $TW$s. $TW_n$ denotes the $n$-th time window which contains $k$ different time points,

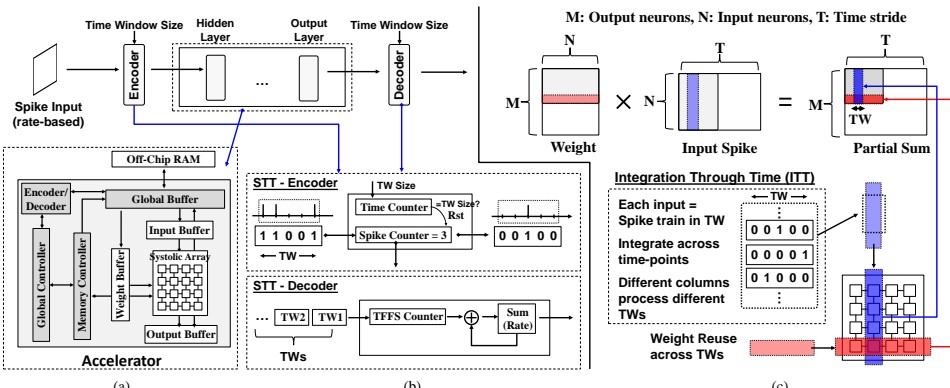

Figure 3: (a): Overall architecture of the proposed accelerator (b): STT-encoder and decoder at the input and output layer, respectively (c): Mapping of the inputs and outputs into the systolic array with the proposed ITT

i.e., $TW_n = (t_{k(n-1)+1}, ..., t_{kn})$. Remaining steps remains the same as in (3) $\sim$ (5) and all the other expressions follows the definition described in (2) $\sim$ (5). Importantly, ITT directly maps the TW to a PE which contains multiple time points which is different from Lee & Li (2020).

## 3.2 MAPPING TO SYSTOLIC ARRAY

We structurize the irregular sparse firing activities with uniformity across TWs based on STT and accelerate input integration steps of multiple TWs in different output neurons in parallel using ITT. With STT and ITT, our mapping strategy enables parallel processing in both 1) time: across multiple time point and 2) space: across different output neurons, which significantly improves data movement and processing time.

The proposed architecture accelerates partitioned matrix-matrix multiplication of the weight and spike input matrices on the systolic array and employs parallelism both across different neurons and different TWs. As shown in Fig. 3(c), PEs in a specific row performs the computations for a particular output neuron across different TWs. In each column, PEs process spike inputs of a given TW for different output neurons. Data are only fed from the edges of the systolic array providing high data distribution bandwidth. In each PE, the PE receives spike input and weight from its upper and left neighbors and passes spike input and weight to its lower and right neighbors.

## 3.3 STT-BASED LAYER ACCELERATION

Each PE accelerates the fundamental operations of a spiking neuron where hardware resources are reused across different steps. Below, we first discuss the processing of feedforward layers followed by that of recurrent layers, for which PE operates with a simple additional step to incorporate recurrent synaptic inputs.

The operations in a single PE follow the three steps (2) $\sim$ (5) with an AC unit and a small scratch-pad shared through the steps, as shown in Fig. 4(a). In Step 1, the synaptic input integration step, the PE determines the address based on the spike timing in a given TW and accumulates the associated weight into a corresponding memory. A single spike in a TW can be interpreted as a one-hot encoded address for the integration. The small scratch-pad memory first stores the integrated synaptic inputs (ISI) of multiple time points in a given TW. In the above operation, a simple combinational logic, one-hot to binary, converts the spike trains of a TW into an address to the small scratch-pad. As shown in Fig. 4(a), for example, if the spike input is 01000 with TW size 5, the associated weight is properly integrated into ISI[TFFS] = ISI[4], which is the integrated synaptic input of the second time point in the TW.

Next, the actual Psum is calculated using the ISI in the previous step. As discussed in Section 2 and shown in Fig. 4(b), we utilize the prefix sum of ISI which restores the rate and temporal information equivalent to a left-aligned rate code counterpart, while sustaining the advantages of using STT with a single spike. As shown in Fig. 4(b), the use of prefix sum yields the same Psum results as using

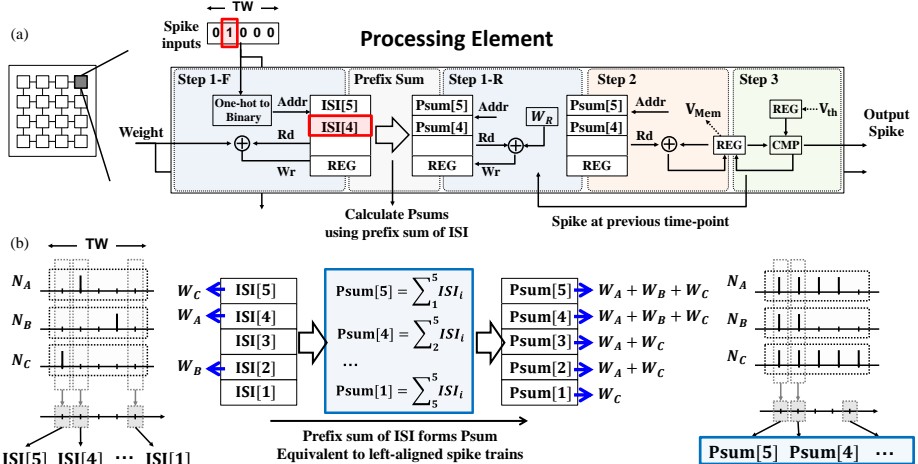

Figure 4: Schematic representations of (a): Operations in a PE for accelerating feedforward and recurrent layer (b): Calculating partial sums (Psums) using a prefix sum of the integrated synaptic inputs (ISI)

left-aligned, rate codes where the rate equals TFFS. Note that, the number of operations to calculate the prefix sum equals to (TW size - 1) which is negligible compared to input integration steps.

For the rest of the operation, PE processes Step 2 and Step 3 with the integrated Psums, time point by time point, in a sequential manner. At a given time point $t_k$, the PE updates the membrane potential with Psum[$t_k$] and the membrane potential of the previous time point $t_{k-1}$. If the updated membrane potential exceeds the pre-defined threshold, the PE generates an output spike and resets the membrane potential.

In case of recurrent layers, the synaptic input integration step is almost the same as that for feedforward acceleration except for one additional step for integrating recurrent synaptic inputs, denoted as Step 1-R in Fig. 4(a). To simplify the recurrent layer processing, we adopt the self-recurrent structure in Zhang & Li (2021) which only requires a single additional integration operation. The proposed PE is capable of accelerating both feedforward and recurrent layers based on the proposed techniques. As will be discussed in Section 4, STT with the use of prefix sum approach delivers competitive accuracy for various networks and significantly improves the accelerator performance.

## 4    RESULTS

We perform comprehensive evaluations of the proposed architecture with various layer types, i.e., fully-connected (FC), convolutional (CONV) and recurrent, focusing on the impact of the proposed STT and ITT following the setups described in Section D. We first examine how the data reuse and computational complexity change upon the proposed techniques with critical architectural parameter, i.e., time window (TW) size. Then, we explore joint optimization of machine learning performance and SNN hardware accelerator performance with application-independent split-time temporal coding. We adopt the state-of-the-art training algorithm proposed in Zhang & Li (2020) as our ML performance baseline, and compare it with our accuracy achieved using the proposed STT over various TW sizes. Since this is the first work of temporal information compression (STT) with time-domain parallel processing (ITT), we set our hardware baseline as the one that has been trained with Zhang & Li (2020) and optimizes data reuse and storage efficiency for each time-point (time-serial approach) without incorporating proposed STT and ITT, as in Khodamoradi et al. (2021); Neil & Liu (2014); Shen et al. (2016).

### 4.1    STT: TEMPORAL INFORMATION COMPRESSION

STT reconstructs the spike information with higher, but structured sparsity by dividing the time stride into multiple TWs, squeezing the entire spike information in each TW to the timing of a single spike. While local spike counts are directly related to the performance of the accelerator,

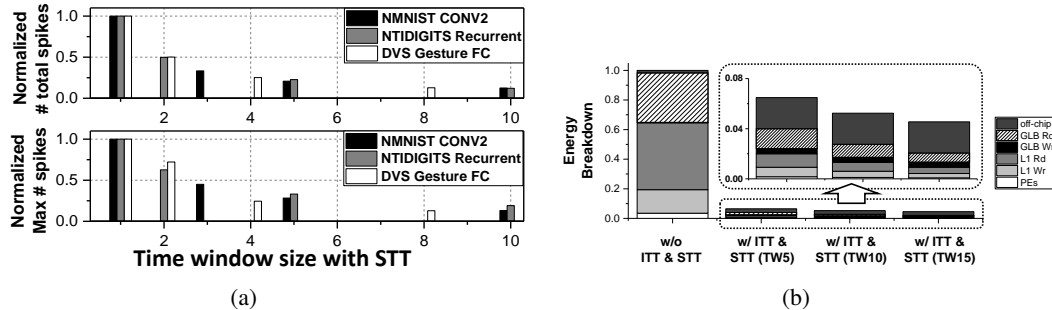

(a)                                 (b)

Figure 5: (a) Normalized number of total spikes and maximum number of spikes in a neuron with different time window sizes. (b) Normalized energy dissipation and energy breakdown with and without the proposed techniques.

having fewer spikes reduces not only the number of accumulate operations but also data movement. As introduced in Section 2, STT applies to all layer types including FC, CONV and recurrent layers. Furthermore, flexibility in TW size selection for STT enables the proposed architecture to accelerate individual applications with different optimizations.

**Computational Overhead:** The number of spikes required for layer acceleration decreases with the TW size, so does the computational overheads by STT. Given the actual spiking activities, the overhead reduction and compression of the spike information differ across layers and networks. Approximately, the number of required AC operations is inversely proportional to the TW size, as shown in Fig. 5(a).

**Data Movement:** STT enables fewer weight data movements associated with active pre-synaptic neurons across the different levels of the memory hierarchy. In conventional approaches, iterative weight access based on the active pre-synaptic neurons at each time point is inevitable due to the sequential processing. However, STT reduces temporal resolution, and more sparsely populated spikes mitigate read and write memory access at each level of memory. For example, the spiking activity of a bursting neuron, which fires across five consecutive time points, forces the integration of the corresponding weight in those five time points repeatedly. This may incur data movements from higher-level caches depending on spiking activities of the pre-synaptic layer and the memory size. In contrast, STT only requires the weight once throughout all the time points in a TW. Data movement and reuse are further improved by the proposed ITT.

## 4.2 ITT: DATA REUSE

ITT significantly improves data reuse by providing data sharing opportunities across TWs and post-synaptic neurons, and minimizes the memory access and stall cycles originating from additional latency for iterative memory access. ITT maps spike inputs in multiple TWs into different columns and enables weight reuse across the PEs in the same row.

We use the recurrent layer trained for the NTIDIGITs as a representative layer to analyze the impact of the proposed techniques in data movements, as shown in Fig. 5(b). Clearly, larger TW sizes reduce access to the higher-level caches and improve energy dissipation. Compared to the conventional approach without the proposed ideas, we observe a huge improvement in memory access to the L1 cache and global buffer. In general, such impact varies from layer to layer based on the actual firing activity, data movements in and between the array and memory hierarchy as a result of the dataflow determined by given memory sizes and layer specifications. Without using the proposed ITT and STT, iterative operations through 300-time points may markedly degrade the overall performance of the accelerator.

## 4.3 COMPREHENSIVE EVALUATIONS

We examine how the proposed STT and ITT with the key architectural parameter TW size improve the overall accelerator performance. Also, we evaluate the tunable tradeoffs between machine learning and accelerator performance in terms of the TW size.

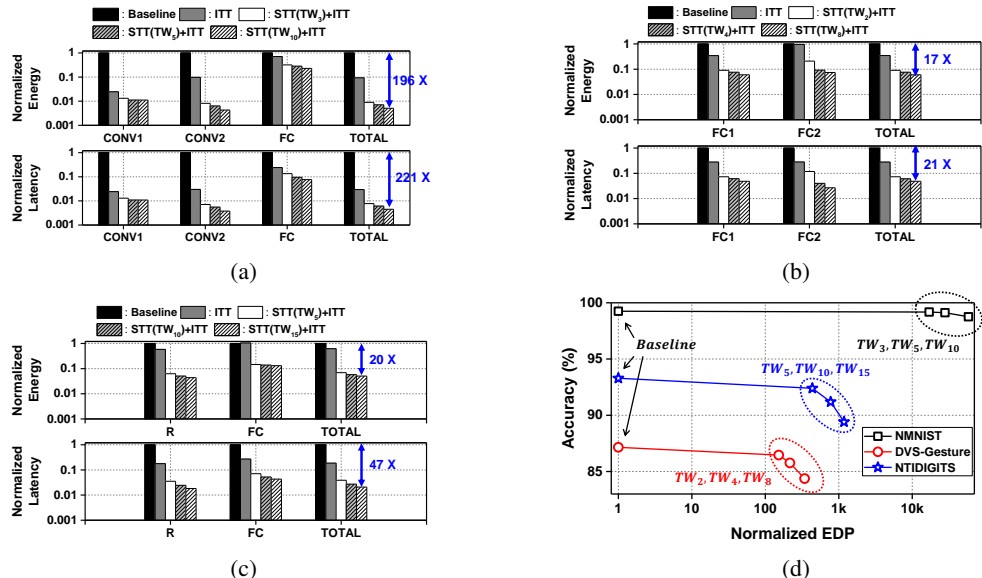

Figure 6: Normalized energy dissipation and latency of layers with different TW sizes for (a): NMNIST, (b): DVS-Gesture, (c): NTIDIGITS, (d): Machine learning performance (inference) - Accelerator performance (normalized EDP) tradeoffs on various datasets.

**Latency:** We observe a huge improvement in latency by using STT and ITT in all three networks, as shown in Figs. 6(a)to 6(c). As discussed in Sections 4.1 and 4.2, 1) STT reduces latency of the computations in the array proportionally to the TW size by processing a TW instead of a time-point, and 2) ITT minimizes the additional delay due to stall cycles resulted from waiting for the required data, by reusing the weight data horizontally. In general, a larger TW size compresses the temporal information with a greater stride in the time domain and further reduces computational overheads and data movements, hence the latency.

However, after a certain TW size, the additional improvement with a much larger TW size decreases. This is due to the fact that spikes are often clustered in a certain range in the time domain as shown in Fig. 1(b), and the number of TWs is the reciprocal of TW size. Also, the impact on latency may vary with the spiking activity depending on how uniformly the spikes spread out through neurons and time points. For example, STT does not reduce the complexity of processing five spikes which are generated by five different neurons. On the other hand, if the five spikes are from a single neuron while other neurons are silent, STT may significantly reduce the computations and weight access. The proposed techniques improved the latency by 97X on average, across the three networks.

**Energy Dissipation:** Energy dissipation is reduced as TW size increases in all layers, similar to the latency. Generally, larger TWs provide the opportunity to reuse the same weight across more time points. Especially, the benefit from data movement/reuse is maximized when the layer has relatively a great amount of weight data, as in CONV2 in NMNIST. In this case, the improvements over the baseline are more pronounced since the baseline exacerbates more data movement and access to higher-level caches due to its iterative weight access.

As discussed, the impact of the proposed techniques on energy dissipation also depends on the temporal sparsity level. For example, a single spike throughout the entire time domain from a particular neuron cannot be reused across TWs and would result in less benefit. Importantly, however, firing activities from a neuron are often clustered in time and in practice weights can be reused through the TWs. Across three different networks, our methods delivered 78X energy dissipation improvement, on average.

**Machine Learning Performance:** Our experimental results present a huge accelerator performance improvement with temporal information compression using STT. However, there exists a fundamental trade-off between accelerator performance and machine learning performance. While STT significantly improves latency and energy dissipation by using structured and sparse spiking activities, STT may cause a local temporal information loss in a TW.

Nevertheless, the STT-based acceleration delivers competitive performance as summarized in Table 1, $TW_i$ denotes that the time window size is $i$. For example, $TW_5 \times 60$ represents that the original spiking activity, spanned through 300-time points, is encoded to 60 consecutive TWs where each TW contains 5-time points with at most a single spike. We adopted the training algorithm in Zhang & Li (2020) and conducted STT-based inference test on well-trained networks with different TW sizes. For example, Zhang & Li (2020) achieved 93.29% accuracy and the STT-based simulation achieved 92.40% inference accuracy with TW size=5 for the NTIDIGITS dataset. We observe that the proposed STT can deliver competitive inference performance up to a certain TW size across various networks as in Table 1 while providing a significant improvements on hardware acceleration.

## 4.4 ML-HW Performance Trade-off

STT significantly reduces computational overhead by introducing local temporal resolution reduction per TW, tunable based on TW size while maintaining global temporal information of the original spikes without complex hyper parameter tuning. Aggressive reduction with larger TW sizes may exacerbate local temporal information loss in TWs, leading to a non-negligible classification accuracy drop albeit that more substantial accelerator performance improvement can be achieved.

We use energy-delay product (EDP) to simultaneously consider latency and energy dissipation for evaluation of the proposed techniques and to analyze the impact of the TW size selection. As shown in Fig. 6(d), the ML-HW performance trade off can be flexibly adjusted depending on application objectives where small TW sizes with STT and ITT can still deliver significant improvement. Our work delivers 15,000X EDP improvement with the maximum TW sizes which do not significantly drop the accuracy, on average across different benchmarks, as shown in Fig. 6(d).

Table 1: Performance on fully-connected, convolutional and recurrent networks: NMNIST, DVS-Gesture and NTIDIG-ITS. TWS denotes the applied time window size.

| Neuromorphic MNIST | | | |
|---|---|---|---|
| Method | Network | Accuracy | Timepoints |
| HM2BP Jin et al. (2018) | 400-400 | 98.88% | 400 |
| SLAYER Shrestha & Orchard (2018) | 500-500 | 98.95% | 300 |
| SLAYER Shrestha & Orchard (2018) | CNN[a] | 99.22% | 300 |
| TSSL-BP Zhang & Li (2020) | CNN[a] | 99.25% | 30 |
| **STT (TWS=3)** | CNN[a] | **99.18%** | $TW_3 \times 10$ |
| **STT (TWS=5)** | CNN[a] | **99.12%** | $TW_5 \times 6$ |
| **STT (TWS=10)** | CNN[a] | **98.76%** | $TW_{10} \times 3$ |
| **STT (TWS=15)** | CNN[a] | **98.10%** | $TW_{15} \times 2$ |

CNN[a]: 12C5-P2-64C5-P2.

| DVS-Gesture | | | |
|---|---|---|---|
| Method | Network | Accuracy | Timepoints |
| RNN He et al. (2020a) | P4-512 | 52.78% | |
| LSTM* He et al. (2020a) | P4-512 | 88.19% | |
| TSSL-BP Zhang & Li (2020) | P4-512 | 87.15% | 300 |
| **STT (TWS=2)** | P4-512 | **86.46%** | $TW_2 \times 150$ |
| **STT (TWS=4)** | P4-512 | **85.76%** | $TW_4 \times 75$ |
| **STT (TWS=8)** | P4-512 | **84.37%** | $TW_8 \times 38$ |

* includes much greater number of tunable parameters.

| N-TIDIGITS | | | |
|---|---|---|---|
| Method | Network | Accuracy | Timepoints |
| HM2BP Jin et al. (2018) | 250-250 | 89.69% | 300 |
| BP (GRU) Anumula et al. (2018) | 200-200-100 | 89.92% | |
| BP (LSTM) Anumula et al. (2018) | 250-250 | 91.25% | |
| TSSL-BP Zhang & Li (2020) | 400[a] | 93.29% | 300 |
| **STT (TWS=5)** | 400[a] | **92.40%** | $TW_5 \times 60$ |
| **STT (TWS=10)** | 400[a] | **91.19%** | $TW_{10} \times 30$ |
| **STT (TWS=15)** | 400[a] | **89.41%** | $TW_{15} \times 20$ |

400[a]: Recurrent layer with LISR Zhang & Li (2021)

## 5 Conclusion

This work is motivated by the lack of efficient architecture for acceleration of irregular firing activities derived from complex spatiotemporal dynamics in SNNs.

The proposed systolic array-based architecture is built upon a novel *Split-Time Temporal coding* (STT) and an *Integration Through Time* (ITT) technique. STT enables structurization and sparsification of the unstructured firing activities on a time window (TW) basis, and ITT further boosts the efficiency of the accelerator with the parallel acceleration of TWs and data reuse in space and time. Our work provides a universally applicable, application-independent solution for the efficient acceleration of the spiking models with the flexibility in choosing TW size. Experimentally, our work delivers 15,000X EDP improvement for various benchmarks, NMNIST, DVS-Gesture and NTIDIGITS, on average compared to the SNN baseline.

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

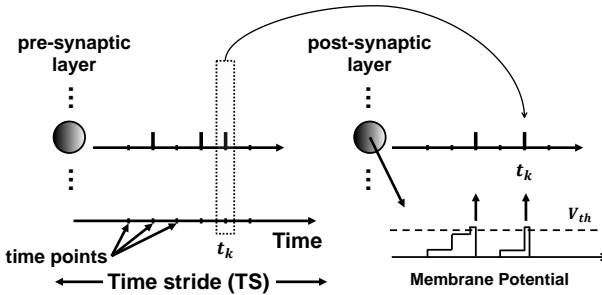

Figure 7: Operations in spiking neural networks (SNNs)

## A BACKGROUND

### A.1 SPIKING NEURAL NETWORKS (SNNS)

Despite the huge success of deep neural networks, SNNs have emerged as a promising alternative with inherent advantages in the event-driven based spatiotemporal data processing. Compared to non-spiking models, operations in SNNs are based on temporal data processing with binary spikes spanning through time which comprise two key distinctions: 1) temporal data processing and 2) data representations.

SNNs are inspired by information processing in the human brain and process information in the time domain over multiple time points. Dynamics in both time and space offer great opportunities to process complex spatiotemporal data but also pose challenges as will be discussed. In this paper, we define *time stride* as a full range of time points that the SNN operates on where *time point* is a minimum unit of time, as shown in Fig. 7.

Another key characteristic of SNNs is data representations. SNNs communicate with binary-valued input/output spikes while weights, membrane potentials and integrated synaptic inputs, i.e., partial sums, are multi-bit. Multi-bit weights are required for the synaptic input integration at each time point, and thus produce key challenges in data movement.

### A.2 FEEDFORWARD SPIKING LAYERS

Conventional and the most natural approach for temporal data processing is to perform operations time point by time point in a sequential manner, for all time points in time stride. In feedforward spiking layers, operations in a single spiking neuron consist of three steps at each time point $t_k$:

**Step 1:** Synaptic input integration at $t_k$:

$$\vec{p}^{Post}[t_k] = \mathbf{W}_{Post,Pre} \times \vec{s}^{Pre}[t_k] \tag{2}$$

**Step 2:** Membrane potential update at $t_k$:

$$\vec{v}^{Post}[t_k] = \vec{v}^{Post}[t_{k-1}] + \vec{p}^{Post}[t_k] - V_{leak}^{Post} \tag{3}$$

**Step 3:** Conditional spike output generation at $t_k$:

$$\vec{s}^{Post}[t_k] = \boldsymbol{f}(\vec{v}^{Post}[t_k]) \tag{4}$$

$$\boldsymbol{f}(v_i^{Post}[t_k]) = \begin{cases} 1, \text{if } v_i^{Post}[t_k] \geq V_{th}^{Post} \to v_i^{Post}[t_k] = 0 \\ 0 \text{else} \to v_i^{Post}[t_k] = v_i^{Post}[t_k] \end{cases} \tag{5}$$

where the *Post* and *Pre* denote the pre-synaptic layer and the post-synaptic layer, and $i$ represents the neuron indices in the post-synaptic layer. $\vec{p}^{Post}[t_k]$, $\vec{v}^{Post}[t_k]$, and $\vec{s}^{Post}[t_k]$ are vectors, representing the integrated partial sum of the spike inputs from the pre-synaptic layer, membrane potential and spike output of the neurons in the post-synaptic layer at time $t_k$, respectively. $\mathbf{W}_{Post,Pre}$ is the

matrix of the feedforward synaptic weights between pre- and post-synaptic layers, $V_{th}$ and $V_{leak}$ are the firing threshold and leaky parameter in post-synaptic layer, respectively. $f$ is a non-linear, all-or-non activation function with a given $V_{th}$. In the above steps, the synaptic input integration (**Step 1**) incurs matrix-vector multiplication and takes place at each time point, comprising the dominant complexity of SNN acceleration.

Importantly, the above steps are repeated at each time point, across all time points in time stride. The above steps present fundamental operations in any feedforward layers including fully-connected and convolutional layers.

### A.3 Recurrent Spiking Layers

Processing neural computations of a recurrent layer in SNNs follow the same three steps in the feedforward layer with additional synaptic inputs. In a recurrent layer, lateral recurrent inputs are also considered in addition to the feedforward input integration (**Step 1**) in (2):

**Step 1\*:** Feedforward synaptic input integration at $t_k$:

$$\vec{p}_F^{Post}[t_k] = \mathbf{W}_{Post,Pre} \times \vec{s}^{Pre}[t_k] \tag{6}$$

$$\vec{p}_R^{Post}[t_k] = \mathbf{W}_{Post,Post} \times \vec{s}^{Post}[t_{k-1}] \tag{7}$$

$$\vec{p}^{Post}[t_k] = \vec{p}_F^{Post}[t_k] + \vec{p}_R^{Post}[t_k] \tag{8}$$

where $\vec{p}_F^{Post}[t_k]$, $\vec{p}_R^{Post}[t_k]$ and $\vec{p}^{Post}[t_k]$ are vectors, representing the partial sum of the feedforward input integration, recurrent input integration, and fully-integrated partial sum in the post-synaptic layer at time $t_k$, respectively. $\mathbf{W}_{Post,Post}$ is the matrix of the recurrent synaptic weights of the post-synaptic (recurrent) layer.

### A.4 Challenges of SNN accelerations

Binary-valued spikes and temporal processing in the time domain open up the opportunities for event-driven processing and support a wide range of spatiotemporal tasks. However, the added temporal dimension introduces crucial challenges in accelerating SNNs: 1) unstructured sparsity in both spatial and temporal domains and 2) iterative weight data access in every time point.

The most natural approach for SNN acceleration, or simply conventional approach in this paper, is to process firing activities time point by time point in a sequential manner, which has been adopted in previous works Cao et al. (2015); Khodamoradi et al. (2021); Cheung et al. (2012); Chuang et al. (2020). Since the operations at each time point are similar to the non-spiking ANN counterpart, the conventional approaches adopt optimized dataflow and mapping strategies for ANNs in essence.

However, sequential processing in time requires unstructured weight access according to the firing activity at the given time point which repeats through all time points in the time stride. For example, the weights required at time point $t_k$ are different from the weights required at the next time point $t_{k+1}$. Alternating weight matrices based on the firing activities cause a high overhead in data movement and significantly degrade the accelerator performance.

## B Overview of the Proposed Architecture

### B.1 Systolic Array

In many prior works, a 2-D systolic array has been adopted as the main computing substrate in accelerating neural networks with clear advantages in complexity, data reuse, locality, data distribution bandwidth and compute density He et al. (2020b); Khodamoradi et al. (2021); Chuang et al. (2020); Kung et al. (2019). While data are fed only from the edges of the array, each data from the top and left propagates vertically and horizontally, i.e., from top to bottom, and left to right, without complicated inter-PE communication. Thus, each processing element (PE) performs the computations with the data from the upper and left neighbor, and all the PEs in the array operates in a synchronized manner. With the advantages in data reuse in vertical and horizontal directions, in particular, we adopt the systolic array as the main computing substrate in this work.

### B.2 PROPOSED ARCHITECTURE

Fig. 3 shows the overall architecture of the proposed architecture incorporating an STT-encoder for the input layer, an STT-decoder for the output layer, controllers, caches, and a systolic array composed of tiled processing elements (PEs) with unidirectional links. As shown in Fig. 3(a), the systolic array fetches the required data through three levels of memory hierarchy: 1) off-chip RAM, 2) a global buffer and 3) double-buffered L1 caches. The received spike input and weight data propagates vertically and horizontally with unidirectional links across the 2-D array and is reused through multiple PEs. Each PE is composed of 1) a simple controller, 2) a small scratch-pad memory, 3) accumulate unit (AC), 4) a simple one-hot-to-binary decoder and 5) a comparator. Unlike multiply-and-accumulate (MAC) operations in non-spiking accelerators, simpler AC units are used to accumulate weight values with binary-valued spikes. To fully leverage STT-based acceleration, the synaptic input is properly integrated into the corresponding time point with a simple decoder, and the scratch-pad in each PE stores the Psums of all time points in a given TW. In the rest of the paper, we primarily focus on synaptic input integration, the dominant computational complexity. Also, detailed advantages of the proposed techniques will be discussed in Section 4 and Appendix C.

## C PERFORMANCE WITH STT

### C.1 MACHINE LEARNING PERFORMANCE

Typically, temporally-coded spiking models limit each neuron to fire at most once in the entire time domain. This highly restrictive type of spike coding may benefit latency and energy efficiency. However, it does not apply to broader classes of SNNs employing rate or other types of temporal codes or a combination of thereof, and limits model accuracy, especially for challenging learning tasks.

On the other hand, rate-coded spiking models can support various types of spatiotemporal dynamics of SNNs. While many recent works based on rate-coded models reported competitive performances on various spatiotemporal tasks with bio-inspired Kheradpisheh et al. (2018); Hao et al. (2020) and backpropagation based Park et al. (2020); Zhang & Li (2020); Jin et al. (2018) training methods, iterative weight access due to repeated operations across time and irregular firing patterns complicate hardware acceleration of the spike-based models.

Importantly, STT is universally applicable to any rate-coded model including fully-connected, convolution and recurrent layers for efficient hardware acceleration of a trained network with flexibility in selecting the temporal granularity, i.e., TW size. STT delivers competitive accuracy without any hyper-parameter tuning and significantly reduces computational overhead for synaptic input integration, the dominant complexity of hardware acceleration. STT is fundamentally different from existing temporal coding schemes Park et al. (2020); Zhang et al. (2019) while the information-carrying feature in a single TW bears a similarity. Results on various networks are discussed in Section 4.

### C.2 HARDWARE PERFORMANCE: ENERGY REDUCTION

The main bottleneck of the SNN accelerators is the data movement/access overhead of multi-bit weight data which is addressed by the proposed techniques. First, STT minimizes the computational overhead required for dense spiking activities with structured sparsification. STT restricts each neuron to fire at most once in a TW and enables the same weight data associated with a presynaptic neuron to be used only once. In general, applying a larger TW size further reduces the computational and data movement overhead with higher sparsity in spiking activities. Data movement/access is further improved with ITT by the improved weight data reuse. PEs in the same row in the array perform computations of a post-synaptic neuron across different TWs, i.e., the same weight data is reused across PEs in the same row with different spike inputs.

Table 2: A high-level overview of the user-defined inputs.

| Input | Description |
|---|---|
| Array configuration | Array width/height, size of the scratch-pad in PE |
| Memory configuration | Size of the memory in three levels: off-chip RAM, Global buffer, L1 cache |
| STT | Use STT-based spiking acitivities or plain counterpart along with time window size |
| ITT | Mapping different TWs across columns of the systolic array with given TW size |
| Time Window ($TW$) Size | Ranging from plain inputs ($TW$=1) to the size of a scratch-pad in PE, i.e., $TW$=50 |
| Layer Type | fully-connected, convolutional and recurrent |
| Network Structure | Number of layers, layer types, and number of the neurons in each layer |

## C.3    HARDWARE PERFORMANCE: UTILIZATION EFFICIENCY AND LATENCY

STT and ITT improve severe under-utilization which originates from iterative data access and the irregularity of sparse firing activities at each time point in the time stride. As discussed in Section 2, each neuron fires at most once in a TW with STT, and thus the processing of any TW takes the same amount of time. Uniformity in processing time across TWs and higher sparsity with STT significantly improve latency and utilization efficiency. Iterative access of the required data at each time point can cause stalls of the array, which is the source of inefficiency in addition to computation latency, while ITT reduces memory access to higher-level caches by the improved weight data reuse and less data movement.

## D    EVALUATION METHODOLOGY

We develop an analytic architecture simulator to support various types of layers, unique characteristics in SNNs, and trace data access/movement for evaluating the latency and energy dissipation for accelerating a specific task. In Table 2, the user-defined inputs for the simulator are summarized.

### D.1    SYSTOLIC-ARRAY AND MEMORY MODELING

#### D.1.1    SYSTOLIC ARRAY

A systolic array is a central computing unit of our simulator and fetches spike inputs and weights from the top and left edges, respectively. The received data propagate vertically and horizontally, and thus each PE grabs the spike input from its upper neighbor and the weights from the left neighbor. In particular, the spike input is a set of binary spikes of a TW and is used as addresses for the accumulate operations in a PE. As in many other works, we use a 128 processing elements (PEs) Chen et al. (2016); Narayanan et al. (2020); Yin et al. (2022) in the systolic array along with double-buffered L1 caches to provide required data to the array. By default, we analyze the architecture performance based on a 16×8 systolic array.

#### D.1.2    MEMORY HIERARCHY

Similar to many other analytic architecture evaluation models, we adopt an off-load model with a three-level memory hierarchy. We follow the standard practice Samajdar et al. (2018); Kwon et al. (2019) to use double-buffering to hide the latency for memory-intensive neural networks and especially separate L1 cache for each type of data, i.e., spike inputs, weights and spike output, for the systolic array operations. The small scratch-pad in each PE stores the Psums of multiple time-points which is similar to the output stationary in non-spiking ANN accelerators. The choice of our architecture is based on many other small-medium scale accelerator works Chen et al. (2016); Narayanan et al. (2020); Shen et al. (2016), and all the memories inside the accelerator are scratchpads hence coherence is not considered. Architecture specifications are summarized in Table 3.

Table 3: Architecture specifications.

| Components | Proposed Architecture |
|---|---|
| Number of PEs | 128 |
| ALU in PEs | Adder, Comparator - 8-bit |
| Global Buffer Size | 54KB |
| L1/Scratchpad Size | 2KB / 50 × 8-bit |
| DRAM Bandwidth | 30GB/sec |
| Bit precisions | Weight/Membrane Potential - 8-bit
Input/Output Spike - $TWS$ × 1-bit
($TWS$: $TW$ size) |

## D.2 PERFORMANCE MODELING

The developed simulator produces dataflow, the procedure to map the computations onto the array, considering the user-defined inputs such as array dimension, number of time-points in time stride and sparsity in actual spiking activities. With the dataflow, the simulator assigns unique addresses for each data and traces read and write in PEs and each level of the memory hierarchy. Following the estimation methods in many previous works Samajdar et al. (2018); Kwon et al. (2019); Peng et al. (2019); Chen et al. (2018); Lee & Li (2020), the simulator calculates latency, memory access and energy dissipation.

### D.2.1 LATENCY

The systolic array performs AC operations through PEs on the array while the required data are continuously accessed from higher- to lower-level cache for stall-free operation. Whenever the required weight and spike input data are ready, the array works for the computation. Thus, the latency is estimated with the worst delay between data access from higher-level cache and computations in the array. The total latency is calculated by adding all latencies across the entire process.

### D.2.2 MEMORY ACCESS

For the given user-defined inputs, the simulator generates a dataflow that pre-determines the data loading onto the array. We use the actual spiking activities from well trained networks to consider more realistic data movement and follow the methodology adopted in Samajdar et al. (2018) but with consideration in distinctive characteristics of spiking models. Based on the data loading schedule, a specific group of data is required for the computations in the array. Therefore, if the data is absent in the lower-level memory, memory access to higher-level memories is required. With the given memory size and data loading schedule, the simulator counts read/write in all levels of memory as in Samajdar et al. (2018); Kwon et al. (2019). For example, when a specific data is required for the array computation but is absent in the L1, it induces global buffer read and L1 write if the data is present in the global buffer.

### D.2.3 ENERGY DISSIPATION

As in many architecture-level evaluation models Kwon et al. (2019); Samajdar et al. (2018); Peng et al. (2019); Chen et al. (2018); Lee & Li (2020), energy dissipation is calculated with the number of memory access at each level of memory and the number of accumulate (AC) operations for accelerating a given task. With CACTI model Muralimanohar et al. (2009) configured for 32nm CMOS technology, energy dissipation in memory is calculated by multiplying the number of memory access and the energy per memory access. Energy dissipation for computations is evaluated by the number of required AC operations Kwon et al. (2019) based on the actual spiking activities in the network.

## D.3 TRAINING ALGORITHM

All the reported machine learning performance are simulated on NVIDIA Titan XP GPU and the implementation of the proposed STT is conducted on Pytorch framework Paszke et al. (2019). We

adopt one of the the state-of-the-art SNN training method Zhang & Li (2020) for training the network. We train networks with the training algorithm proposed in Zhang & Li (2020) and evaluate the architecture performances using the actual spiking activity data with or without the proposed STT based on different TW sizes, extracted from the well-trained models. To primarily focus on the proposed techniques, we adopt self recurrent structures in Zhang & Li (2021) for the recurrent layers.

## D.4    BENCHMARKS

The proposed STT and ITT are evaluated on various image and speech tasks including neuromorphic image dataset N-MNIST Orchard et al. (2015), neuromorphic video dataset DVS-Gesture Amir et al. (2017), and neuromorphic speech dataset N-TIDIGITS Anumula et al. (2018) with various layer types, i.e., fully connected, convolutional and recurrent, in the network.

**NMNIST** is a spiking version of MNIST dataset. Each sample is a spatio-temporal pattern with 34×34×2 spike sequences of 300ms.

**DVS-Gesture** consists of 1,463 test samples with 11 different classes of hand gestures, where gestures are recorded by a dynamic vision sensor (DVS) camera and converted into neuromorphic data.

**N-TIDIGITS** is the neuromorphic version of the speech dataset, Tidigits speech corpus Leonard & Doddington (1993), where the original audios are converted into spike inputs by a 64-channel CochleaAMS1b sensor.

