# OpenReview forum: "Systolic Array Acceleration of Spiking Neural Networks with Application-Independent Split-Time Temporal Coding"
_ICLR.cc/2024/Conference — Submitted to ICLR 2024_

### Official Review · Reviewer_5XJK · 2023-10-27

**Soundness:** 2 fair
**Presentation:** 3 good
**Contribution:** 2 fair
**Rating:** 5
**Confidence:** 4

**Summary:**

This paper proposes a technique and architecture that allows the exploitation of temporal information compression with structured sparsity and parallelism across time. an Integration Through the proposed Time (ITT) technique that processes integration steps across different TWs in parallel with a systolic array. Experiments showed that the proposed method could deliver 77X and 60X latency and energy efficiency improvements when conducted on different benchmarks.

**Strengths:**

1.This paper is well-written.
2.Application-Independent Split-Time Temporal Coding is useful in reducing time latency when used SNN models. Through the proposed split way, the model could be energy efficient and better performance than other models.

**Weaknesses:**

1. The network architecture is more complex so the energy efficiency could not be utilized fully.
2. When conducting large-scale networks on different benchmark datasets, only MNIST, DVS-Gesture and N-tidigits are small to present the advantages of the model accurately.
3. What is the meaning of timepoints? What is the difference between it and timesteps?
4. Please give more details about energy-delay product (EDP).

**Questions:**

Overall, this paper just wants to split time temporal coding method to reduce the time latency in SNNs. The idea is good, but the whole network structures and training methods are confusing to me. The authors must conduct the proposed method on more large-scale datasets such as imagenet which could fully demonstrate the advantages of the proposed model.

---

### Official Review · Reviewer_iKT8 · 2023-10-31

**Soundness:** 2 fair
**Presentation:** 1 poor
**Contribution:** 2 fair
**Rating:** 5
**Confidence:** 2

**Summary:**

The authors describe a hardware architecture to accelerate the
simulation of spiking neural networks which is based on simplifying
the spike train to essentially a local rate-code (called STT). In STT,
the amount of spikes (and not the exact time) is coded in the
position of the spike in the window, with the additional constrain
that only one spike can occur. They further define a detailed
architecture and show how the proposed
hardware architecture can efficiently compute the required synaptic
integration using the STT code. In benchmarks, they show the trade-off
in accuracy when using the STT with larger time windows.

**Strengths:**

Overall, it is an interesting approach to accelerating SNN simulation
and they show impressive speed-ups with small accuracy drops for
common SNN benchmarks.

**Weaknesses:**

* Spike-time information is dropped
within the time window, thus reducing accuracy. Also, one could simply
use a local spike rate approach (see below) to improve the speed of
the implementation for standard compute (GPUs) as well, so that the
reported runtime increase will likely be much less impressive.

* The presentation is sometimes hard to follow since many of the details
are hidden away in the appendix. For instance, the variables and
notations of EQ 1 are only defined in the appendix. Also what EDP or
PE is, is not clear from the main text. In general, it seems that the
9 pages are not enough to fully describe the architectural details of
the proposed hardware.

**Questions:**

What is missing from the discussion is an obvious alternative to the
STT: instead of position in the time window (TW) coding for the number of
spikes, one could simply count the number of spikes in the TW and
represent it with an integer number (the local spike count in TW) $0 \le k \le n$ where $n$ is the length of the
window. Then one could simply compute once $W\mathbf{k}$ for complete synaptic integration in TW. This would
similarly reduce the weight-reuse dramatically and would not need to
allocate different scales for the various time steps within the TW. Since spike-time
information is eliminated within a TW when using STT, this looks to me an equivalent
approach. Moreover, it seems that the implementation might be much simpler and
energy savings considerable. However, the systolic array architecture
might not support an integer multiplication. It would be interesting to discuss this alternative approach.

---

### Official Review · Reviewer_mhhP · 2023-11-01

**Soundness:** 2 fair
**Presentation:** 3 good
**Contribution:** 2 fair
**Rating:** 5
**Confidence:** 3

**Summary:**

This paper is pinned on computing architecture for accelerating SNNs. The whole study is dedicated to deployment on the systolic array. The Split-Time Temporal coding (STT) techniques split spikes into time windowes (TW) and perform a rate-to-first-spike-time conversion within TW, ensuring at most a single spike presented in any individual TW. The Integration Through Time (ITT) partitions and processes the data parallelly according to TW while reusing/sharing weights across TWs.

**Strengths:**

From the perspective of the EDP metric, this work achieves highly optimized efficiency compared to baseline implementation of SNNs inference. The performance loss of TSSL-BP is relatively acceptable considering the efficiency brought by the acceleration.

**Weaknesses:**

1. A first concern is the hardware baseline. The authors measure EDP against the hardware baseline without STT or ITT. However, parallel processing SNNs in terms of TW has been proposed in [1], which is similar to the ITT methods. I think the authors should consider some recent acceleration baselines. Besides, the difference between ITT and [1] should be discussed since both introduce parallel computing towards time-windowed data.
2. The STT-based compression actually reorganizes the spikes within all TWs by putting them to the end of the window (conversion to first spike time and doing prefix-sum), which breaks the order of spikes inside TW. Since Table 1 does not expose severe performance degradation when TWS grows larger, the reason could also be that the datasets themselves are insensitive to such reorganization of spikes. In such cases, the low-performance loss should be ascribed to the property of datasets rather than the proposed methods.
3. The authors claim TSSL-BP to be a state-of-the-art SNN training method, while seas of novel studies with notably higher training accuracies, such as IM-Loss[2], TEBN[3], TIT[4], based on surrogate gradient has been proposed in the past few years. Since the STT only relies on the rate-based coding scheme which is shared among most nowadays SNNs, the authors should also perform similar experiments on surrogate gradient-trained SNNs.


[1] Jeong-Jun Lee, Wenrui Zhang, and Peng Li. Parallel time batching: Systolic-array acceleration of sparse spiking neural computation. HPCA. 2022.
[2] Yufei Guo, et al. IM-loss: information maximization loss for spiking neural networks. NeurIPS 2022.
[3] Chaoteng Duan, et al. Temporal effective batch normalization in spiking neural networks. NeurIPS 2022.
[4] Shikuang Deng, et al. Temporal efficient training of spiking neural network via gradient re-weighting. ICLR 2022.

**Questions:**

1. Could author point out and discuss the differences between ITT and parallel computing methods proposed in [1]?
2. Why does reorganization of spikes within TW bring very few accuracy loss? Could it might be that there are relatively poor temporal information in these datasets?

---

### Official Review · Reviewer_MbsR · 2023-11-01

**Soundness:** 3 good
**Presentation:** 3 good
**Contribution:** 2 fair
**Rating:** 3
**Confidence:** 4

**Summary:**

This paper presents a temporal compression method that introduces a structural sparsity to the rate coding for spiking neural networks (SNNs). The compression method is called Split-Time Temporal coding (STT) which limits the number of spikes within a time window. This work also proposes a hardware accelerator that exploits STT to reduce data movement, and consequently latency.

**Strengths:**

-- The proposed method is effective in reducing the the spikes within a time window.
-- The structure of the proposed accelerator has been explained clearly.
-- There is a significant reduction in terms of latency.

**Weaknesses:**

-- No comparison was provided with other accelerators for SNNs.
-- There is no information on the main characteristics of the hardware accelerator such as its CMOS technology, power consumption, frequency, area, memory and etc.
-- There is a performance degradation when using the proposed coding.
-- The datasets used in this work are not challenging. The result of more challenging datasets such as ImageNet or CIFAR100 should be included.

**Questions:**

Are the conventional approach and STT supposed to generate a similar output in Fig. 2(b)? If so, how does STT compensate for the inaccurate weighed sum? Is it the source of error and the accuracy degradation?

---

### Official Review · Reviewer_3xhW · 2023-11-02

**Soundness:** 1 poor
**Presentation:** 2 fair
**Contribution:** 1 poor
**Rating:** 3
**Confidence:** 4

**Summary:**

The paper suggests several optimization techniques to sidestep known bottlenecks on spiking neural network hardware accelerators. Specifically, the authors propose Split-Time Temporal Coding, which increases sparsity and Integration Through Time scheme, both operating on time windows that can be processed in parallel. The authors report a latency reduction of 77x and improved energy efficiency by 60x on standard benchmarks such as DVS gesture.

**Strengths:**

The manuscript proposes improvement strategies for SNN simulation efficiency on digital hardware, which is currently one of the primary bottlenecks for SNN research and widespread use in applications and, thus, an important research direction.

The authors propose concrete coding strategies that reduce the computational overhead through increased sparsity and thus reduced data movement.

**Weaknesses:**

The advantages of the proposed coding strategy did not convince me. The Split-Time Temporal Coding targets spiking networks that extensively use rate-coding neurons and are not optimized for sparsity. It is well documented that spiking neural networks, which primarily rely on rate coding neurons, are less efficient than ANNs when simulated on digital hardware (Davidson and Furber 2021). By not explicitly considering this during training, the authors may be solving a problem of their own making.

The methods were incomplete and opaque, and it was not clear how the networks were trained. For instance, basic information as to how input was encoded or how many time-steps were used for simulating the network was missing? Was sparsity encouraged during training? The authors point to Zhang & Li (2020), but in their work, they often use only five time steps, which many would not consider a spiking neural network. Overall, it was unclear what the proposed strategies gain over established state-of-the-art work.

The presentation of the results took a lot of work to follow. For instance, I could not find the 77x latency reduction proclaimed in the abstract in the results figures.

One misses a thorough comparison to existing work that proposed similar yet more powerful coding schemes, e.g., (Stöckl and Maass 2021).

The present work mainly compares to its reference implementation. For instance, one misses a comparison to job that tried to quantify the computational cost more carefully, e.g. (Yin, Corradi, and Bohté 2021).



## References

Davidson, Simon, and Steve B. Furber. 2021. ‘Comparison of Artificial and Spiking Neural Networks on Digital Hardware’. Frontiers in Neuroscience 15. https://doi.org/10.3389/fnins.2021.651141.

Stöckl, Christoph, and Wolfgang Maass. 2021. ‘Optimized Spiking Neurons Can Classify Images with High Accuracy through Temporal Coding with Two Spikes’. Nature Machine Intelligence 3 (3): 230–38. https://doi.org/10.1038/s42256-021-00311-4.

Yin, Bojian, Federico Corradi, and Sander M. Bohté. 2021. ‘Accurate and Efficient Time-Domain Classification with Adaptive Spiking Recurrent Neural Networks’. Nature Machine Intelligence 3 (10): 905–13.

**Questions:**

How does the propose algorithm compare to existing benchmarks which usually use eFLOPS or binary OPs, e.g., Yin, Corradi, and Bohté (2021)?

How does the proposed work compare to existing benchmarks that directly measure energy consumption, e.g., Blouw et al. (2019).

What are the savings of STT coding if the networks are explicitly optimized for sparsity as was, for instance, done by Cramer et al. (2022)?

What are the weaknesses, i.e., when does the proposed coding and integration scheme work and in which situations does it break down?


## References

Blouw, Peter, Xuan Choo, Eric Hunsberger, and Chris Eliasmith. 2019. ‘Benchmarking Keyword Spotting Efficiency on Neuromorphic Hardware’. In Proceedings of the 7th Annual Neuro-Inspired Computational Elements Workshop, 1–8. NICE ’19. Albany, NY, USA: Association for Computing Machinery. https://doi.org/10.1145/3320288.3320304.

Cramer, Benjamin, Sebastian Billaudelle, Simeon Kanya, Aron Leibfried, Andreas Grübl, Vitali Karasenko, Christian Pehle, et al. 2022. ‘Surrogate Gradients for Analog Neuromorphic Computing’. Proceedings of the National Academy of Sciences 119 (4). https://doi.org/10.1073/pnas.2109194119.

---

### Meta-Review · Area_Chair_kMnD · 2023-12-03

**Metareview:**

The paper suggests several optimization techniques to improve the simulation of spiking neural networks
sidestep known bottlenecks on spiking neural network hardware accelerators. However, the reviewers had multiple concerns with the paper, both w.r.t. to its results and its presentation, and there was a clear support for not accepting it.

**Justification For Why Not Higher Score:**

Clear consensus.

**Justification For Why Not Lower Score:**

nan

---

### Decision · Program_Chairs · 2024-01-16

Reject